



# Simulating permeability reduction by clay mineral nanopores in a tight sandstone by combining µXCT and FIB-SEM imaging

Arne Jacob[1], Markus Peltz[2], Sina Hale[3], Frieder Enzmann[1], Olga Moravcova[1], Laurence N. Warr[2], Georg Grathoff[2], Philipp Blum[3] and Michael Kersten[1]

[1]Geosciences Institute, Johannes Gutenberg-University, J.-J. Becherweg 21, 55099 Mainz, Germany
[2]Institute of Geography and Geology, University Greifswald, Friedrich-Ludwig-Jahn-Str. 17a, 17487 Greifswald, Germany
[3]Institute of Applied Geosciences (AGW), Karlsruhe Institute of Technology (KIT), Kaiserstraße 12, 76131 Karlsruhe, Germany

*Correspondence to:* Arne Jacob (a.jacob@uni-mainz.de)

## Abstract

Computer microtomography (µXCT) represents a powerful tool for investigating the physical properties of porous rocks. While calculated porosities determined by this method typically match experimental measurements, computed permeabilities are often overestimated by more than one order of magnitude. This effect increases towards smaller pore sizes, as shown in

this study, in which nanostructural features related to clay minerals reduce the permeability of tight reservoir sandstone samples. FIB-SEM tomography was applied to determine the permeability effects of illites at the nanometre scale and Navier-Stokes-equations were applied to calculate the permeability of these domains. With this data, microporous domains (porous voxels) were defined using microtomography images of a tight reservoir sample. The distribution of these domains could be extrapolated by calibration against size distributions measured in FIB-SEM images. For this, we assumed a mean

permeability for the dominant clay mineral (illite) in the rock and assigned it to the microporous domains within the structure. The results prove the applicability of our novel approach by combining FIB-SEM with X-ray tomographic rock core scans to achieve a good correspondence between measured and simulated permeabilities. This methodology results in a more accurate representation of reservoir rock permeability in comparison to that estimated purely based on µXCT images.

## 1 Introduction

Depositional environment and subsequent diagenetic alterations are two key factors that influence the bulk mineralogical composition and the authigenic clay mineral inventory of a reservoir (Wilson and Pittman, 1977; Worden and Morad, 1999), and therefore the fluid flow properties of the porous rock. A well-established technique to image and analyse rapidly the 3D





physical properties of porous rocks is computer X-ray microtomography (µXCT) combined with the concept of Digital Rock

Physics (Andrä et al. 2013a, 2013b; Okabe and Blunt, 2004). By applying monochromatic synchrotron radiation, it is

possible to overcome conventional µXCT artefacts like beam hardening and problems that arise due to limited phase

contrast, lack in resolution and edge preservation, as well as low signal-to-noise ratios (Brunke et al. 2008; Kling et al. 2018;

Lindquist et al. 2000; Mayo et al. 2015; Spanne et al. 1994). Synchrotron based µXCT images with voxel resolutions in the

order of 1 µm can provide a sound basis for flow and transport modelling of tight sandstones as suggested by Peng et al.

(2014). They found that synchrotron µXCT imaging is necessary for tight sandstones when the connectivity of the pore

space is low and pore throats cannot be resolved using a conventional µXCT scanner. They further concluded that a high

abundance of the smallest resolvable pores falsifies modelled permeabilities due to an overestimation of actual pore sizes.

Several studies have shown sub-micrometre pore structures to be a frequent feature of tight reservoir rocks (Jiang, 2012;

Shah et al. 2016; Soulaine et al. 2016). Most of these nanostructures are related to different types of clay minerals; most

commonly illite, kaolinite, chlorite and smectite (e.g. Wilson and Pittman, 1977,  Worden and Morad, 1999, Desbois et al.

2016). Although known for decades, considering such structural features below µXCT resolution in pore-scale models

remains challenging (Alyafei et al. 2015; Guan et al. 2019; Menke et al. 2019; Peng et al. 2012). Soulaine et al. (2016)

systematically analysed the effect of sub-resolution domains with varying permeabilities on the simulated permeabilities of

Berea sandstone (20 vol.-% porosity, 2 vol.-% sub-resolution domains) and found that calculated permeabilities can be

reduced by up to 50 %, if microporous domain permeabilities converge towards zero. Thus, it is evident that neglecting sub-

resolution information can lead to a significant overestimation of rock permeability in such simulations (e.g. Saxena et al.

2018, 2017). Menke et al. (2019) utilized a multi-scale Brinkman area approach applying different permeabilities for each

microporous domain to simulate flow in mono-mineralic carbonate rock. They showed that Stokes-Brinkman models are in

good agreement with experimental data whereas Stokes and/or Navier-Stokes models alone were not able to predict

permeability in a conventional flow scenario. They also demonstrated that for pure carbonates, a direct correlation can be

established between observed density contrasts and specific physical properties, such as porosity and permeability. However,

this approach is not applicable in a system with more than one rock forming mineral, such as a tight sandstone, where

density contrasts relate to different mineral phases as well as sub-resolution porosities.



The lack of distinct material information for a voxel is often ascribed to as the *partial volume effect* (e.g. Kessler et al. 1984; Ketcham and Carlson, 2001). To overcome this issue, imaging techniques that can resolve the pore structure at different

length scales have to be applied. For estimating the permeability of reservoir rocks, the resolution achieved by synchrotron radiation imaging lies within an acceptable range (Saxena et al. 2018). Several studies have demonstrated that, by combining X-ray and scanning electron imaging, the pore space of tight clay-bearing rocks can be spatially resolved from the mm-down to the nm-scale (e.g. Desbois et al. 2016; Hemes et al. 2015; Markussen et al. 2019).

In this study, we aim to demonstrate a new approach by combining synchrotron-based μXCT imaging with focussed-ion-

beam scanning electron microscopy (FIB-SEM) to improve flow simulations in a tight sandstone formation with high clay mineral content. First, we used machine learning-based image segmentation to enhance pore space segmentations of artefact rich FIB-SEM topologies. Then, we conducted Navier-Stokes simulations on FIB-SEM topologies. Finally, we subsequently used these simulations as input data for sub-resolution domains in μXCT based Stokes-Brinkman models (Brinkman, 1949; Neale and Nader, 1974). This novel morphology-based approach for sub-resolution-rich materials results in simulated

permeabilities that fit experimental results significantly better than using Navier-Stokes simulations alone.





## 2 Sample preparation and characterization

In this study, a well-characterized illite-bearing Upper Rotliegend sandstone from Germany was used, which was sampled from the quarry Schwentesius near the Bebertal village (Heidsiek et al. 2020). This well-known location exposes an analogue of the Permian gas reservoir sandstone of the Flechtingen High, which formed part of the North German Basin.

Thin section analyses of this reservoir rock have shown that illite is the main clay mineral, which primarily occurs as a coating along detrital grains, as illite meshworks grown on coatings, as diagenetically altered K-feldspar and as illitized lithoclasts (Fischer et al. 2012).

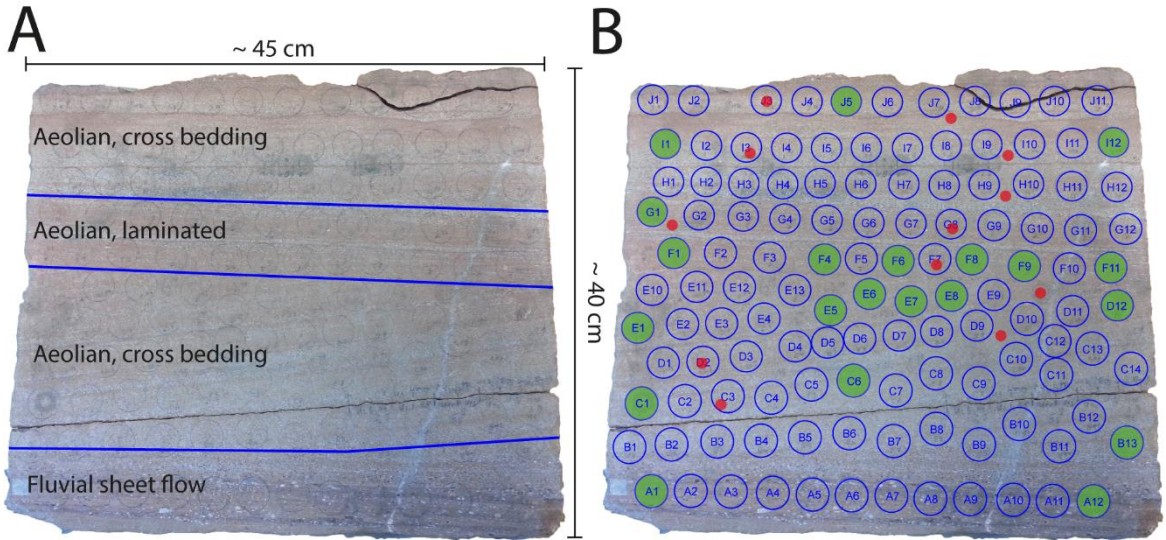

**Figure 1 (A) Upper Rotliegend sandstone block showing four main deposition facies. (B) Sampling locations and sizes**
**of the extracted plugs (green) and the mini plugs (red) extracted.**

Samples were taken from a large sandstone block displaying a variety of different aeolian and fluvial depositional facies (Fig. 1A). We identified several facies denoted A to J from four different types of sedimentary depositional layers. The samples of the different facies were numbered from left to right. Samples were drilled out and extracted from marked locations in the form of plugs with a diameter of 2.5 cm. The plugs were cut into three segments which were used for X-ray

diffraction analyses, helium porosimetry measurements and FIB-SEM imaging. Mini plugs were drilled directly beside the plugs with a diameter of 3 mm and a length between 10 and 20 mm. The larger plugs were used for mineralogical/geochemical analysis, and to measure permeabilities experimentally. The mini plugs were used for the

synchrotron radiation-based µXCT imaging at the PETRA beamline P05 of DESY Hamburg (Germany). After the

synchrotron measuring campaign, the mini plugs were additionally examined by FIB-SEM and Energy Dispersive X-ray

Spectroscopy (EDX) imaging to obtain qualitative and quantitative information about the clay mineral particles found within

the rock pore space. Microporous structures in the Rotliegend sandstone sample could be resolved by comparing µXCT and

FIB-SEM images. The term "microporous" refers to the definition of sub micrometre porosity by Soulaine et al. (2016), who

differentiated between void, solid, and microporous voxels in µXCT images. "Nanoporous" is used to describe structures

with predominant pore sizes in the nanometre range (0.2 – 1000 nm).

**3 Analytical procedures**

**3.1 Mineralogical characterization**

X-ray diffraction analyses using the Rietveld analysis program Profex 4.0 (Doebelin and Kleeberg, 2015; Ufer et al. 2012)

have shown a homogenous mineralogical composition along the layers of the sample block with only slight variations in

content (Figure A1). The main components are quartz (58-69 wt.-%), authigenic and diagenetic feldspars (12-20 wt.-%),

calcite (1-18 wt.-%), and illite (10-17 wt.-%). In the <0.2 µm size fraction, we observed traces of swelling smectite

interpreted as contaminants from surface weathering. Accessory hematite and barite particles also occur with abundances of

<1 wt.-%. In the Bebertal tight sandstone, illite is by far the most abundant clay mineral and makes up ~95 wt.-% of the <2

µm fraction. Commonly, the nanometre-scale microstructural features of reservoir rocks containing significant amounts of

clay minerals are usually not detectable at µXCT resolution due to a low absorption contrast (Ahmad et al. 2018).

**3.2 Permeability measurements and pore size distribution**

A helium gas-driven permeameter under steady-state conditions was used to experimentally obtain permeability values for

the individual plugs (e.g. Filomena et al. 2014). Compressed air was used to apply a pressure of 10 bar to the core samples

that were coated by a latex membrane. For permeability measurements, the inflow and outflow pressures of the helium flux

were sequentially increased in up to six pressure steps. The differential pressure was kept constant at 200 – 500 mbar

depending on the sample properties. The intrinsic sample permeability was derived from the apparent gas permeability, $K_g$,

determined for each pressure step using Darcy´s law (Liu et al. 2017):



$$K_g = \frac{2Q p_2 \eta L}{A(p_1{}^2 - p_2{}^2)} \qquad (1)$$

where $Q$ is the measured gas flow rate, $\eta$ is the dynamic viscosity of the permeant, $L$ is the sample length, $A$ is the sample cross-section, and $p_1$ and $p_2$ are the inflow and outflow pressures. By plotting $K_g$ against $\frac{1}{(p_1+p_2)/2}$, the data can be fitted by

a straight line. The intercept of the best-fit line at the $K_g$ axis corresponds to the intrinsic sample permeability, $K_{int}$ (Gao and Li, 2016; Klinkenberg, 1941). Also, MIP measurements were conducted with an Autopore IV Series (Micromeritic Instrument Corp.) to determine the pore size distribution of a dried sub-sample with a weight of ~2.5 g, which was taken from a cross-bedded aeolian layer of the sandstone block. Based on the capillary law, MIP enables the analysis of a wide spectrum of pore sizes (3 nm to > 900 µm), corresponding to a pressure range of 0 - 414 MPa. As a non-wetting liquid with a

high contact angle (130 - 140°), mercury only penetrates a pore when pressure is applied. Under the assumption of cylindrical pores, the applied pressure is directly proportional to the pore throat diameter as described by the Washburn equation (Washburn, 1921):

$$D = \frac{-4\gamma \cdot \cos\theta}{P} \qquad (2)$$

where $D$ is pore throat diameter, $\gamma$ the surface tension, $\theta$ the contact angle, and $P$ the applied pressure.

**3.3 Synchrotron-based µXCT**

For synchrotron tomography, a beam energy of 29.87 KeV was used. The effective image resolution of the detector equipped with a CCD camera was 1.22 µm per pixel, while the image size was 3056 × 3056 pixels. We used an advanced reconstruction script with the MATLAB® software and binned the images by a factor of 2 before reconstruction to increase the signal-to-noise ratio (Moosmann et al. 2014). This decreased the resolution to 2.43 µm and changed the image size to

1528 × 1528 pixels. The number of projections was 1200, with the information of five subsequent images used to calculate an average for every projection image. After reconstruction of the 3D image stacks, the scans were denoised using the non-local means filter of the *GeoDict 2020* software package (Buades et al. 2011). Image segmentation of the mini plugs was realized by conventional greyscale thresholding. A comparison with machine learning segmentation methods revealed a better pore-to-solid segmentation and resolving of small pore throats by thresholding (Fig. 2). Since the main goal was to





achieve the best possible permeability estimation, the differentiation between pore and solid is more important for

permeability estimation than the accurate segmentation into different phases (Khan et al. 2016). Leu et al. (2014) point out

that even a small variation in pore throat morphology can have a large impact on the estimation of permeability.

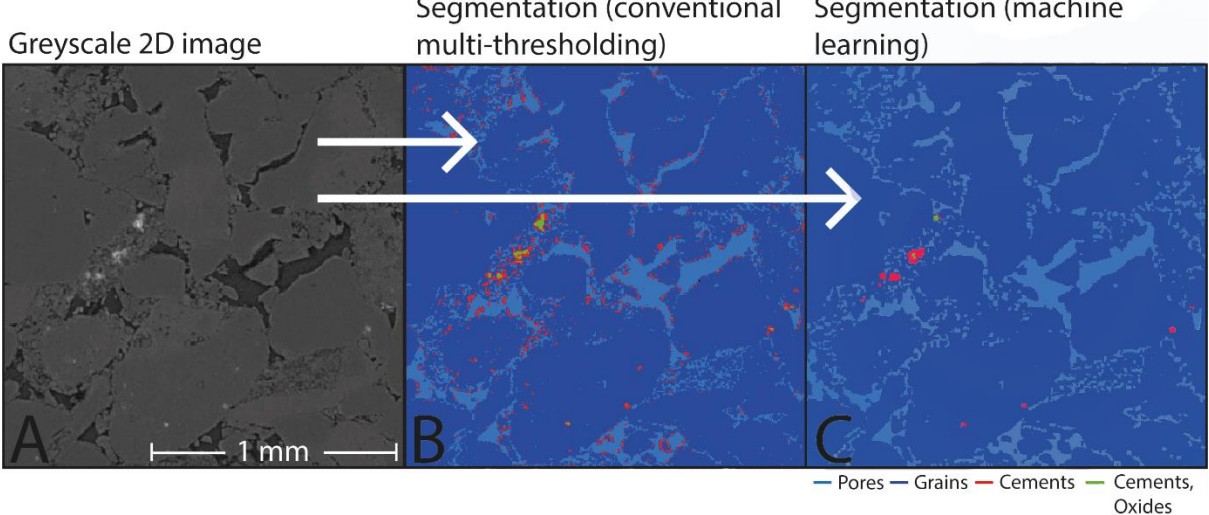

**Figure 2 (A) Greyscale 2D μXCT image of a mini plug. (B) Segmented phases using multi thresholding. (C)**
**Segmented phases using the machine learning image classification module of the software *ilastik* (Version 1.3.3) by**
**Berg et al. (2019)**

For permeability simulations, we either used the Fast Fourier Transformation (SIMPLE-FFT) or the Left-Identity-Right

(LIR) solvers, both implemented in the *FlowDict* module of the *GeoDict* software package (Linden et al. 2015; Moulinec

and Suquet, 1995). While the SIMPLE-FFT solver is fast for calculating low porosity domains, the LIR solver is better

suited for high porosity domains and requires less memory. Both iterative finite volume solvers can apply Navier-Stokes and

Navier-Stokes-Brinkman equations. The equations are derived from Darcy's law (Eq. 1) to calculate the permeability of a

material (Darcy, 1856):

$$\vec{u} = -\frac{K}{\eta}(\nabla p - \vec{f}) \tag{3}$$

In Eq. 3, $\vec{u}$ is the 3-dimensional average fluid-flow velocity, $K$ the permeability, $\eta$ the fluid viscosity, $p$ the intrinsic average

pressure tensor, and $f$ the force density field, which was defined using the Navier–Stokes conservation of momentum

equation for all three dimensions (Eq. 4):

$$-\eta\Delta\vec{u} + (\rho\vec{u} \cdot \nabla)\vec{u} + \nabla p = \vec{f} \tag{4}$$





The Brinkman term can be added to the Navier-Stokes equation where porous voxels are required. These voxels include the nanoporous flow resistivity:

$$-\eta\Delta\vec{u} + (\rho\vec{u}\cdot\nabla)\vec{u} + \eta K^{-1}\vec{u} + \nabla p = \vec{f} \tag{5}$$


where $K^{-1}$ is the inverse of the permeability tensor and $\eta K^{-1}$ the flow resistiveity. The applicability and robustness of combining Navier-Stokes equations with the Brinkman term has been validated by Iliev and Laptev (2004).

We calculated the permeability with symmetric boundary conditions in tangential and flow direction with a pressure drop of 20370 Pa. The symmetric boundary conditions are valid for low porous structures with non-periodic pore throat geometries.

The differential pressure value was set to be able to compare the results with helium permeation flux measurements where similar values have been used. As a convergence stopping criterion, an error bound was used. This stops the iteration when the relative difference to a predicted permeability within the last 100 iterations is smaller than 5 %.



### 3.4 FIB-SEM Measurements

In this study, a Zeiss Auriga crossbeam electron microscope equipped with a Gemini electron column and an Orsay Physics

ion beam was used. SEM images were taken at 1 kV with an in-lens secondary electron (SE2) detector, and FIB slicing was

executed with a beam current between 0.5 and 2 nA and a voltage of 30 kV. This resulted in a slice thickness of 25 nm. A

large FOV of ~20 µm could be reached. To derive structural information from the FIB-SEM images, extensive post-

processing of the data was required. Following image alignment and cropping, stripes and shadow artefacts were filtered out

before image segmentation. The *Slice Alignment* operation of the module *ImportGeo* of the *GeoDict 2020* software package

was used to align the images, while the *Curtaining Filter* was used for stripes correction (Fig. 3). In general, the

segmentation of pores in FIB-SEM images was not straight forward since scans of porous polished sections are pseudo-2D

and contain information from behind the actual imaging plane (De Boever et al. 2015).



FIB-SEM image processing (workflow)

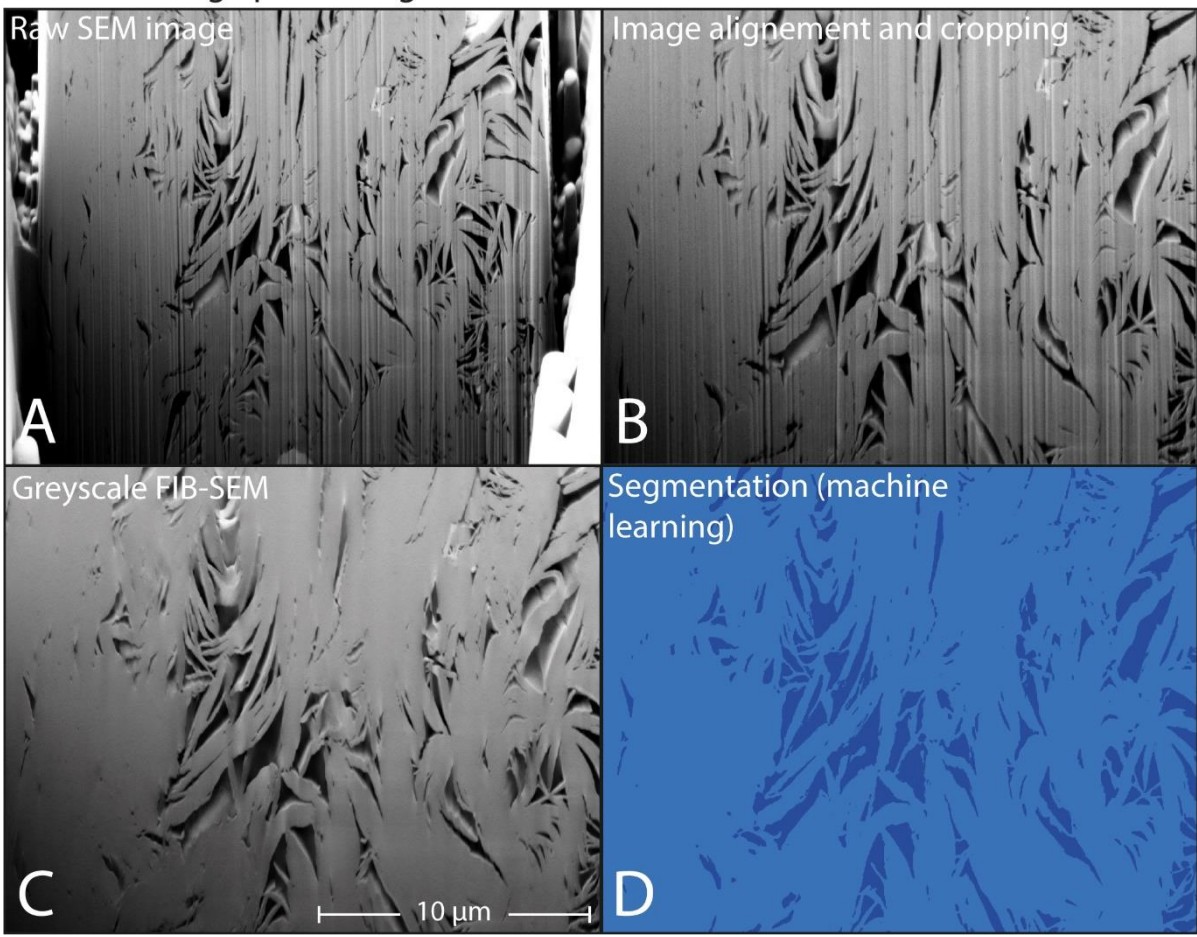

**Figure 3 (A) Raw SEM image of secondary illite growth in a porous feldspar with streaks and shadow artefacts. (B)**
**Images with aligned slices of a cropped region of interest. (C) Filtered greyscale SEM image of the illite meshworks.**
**(D) The binarized result after image segmentation with pixel classification algorithms of the software *ilastik***
As multi-thresholding and watershed segmentation algorithms have problems with shine through artefacts (Prill et al. 2013), capturing the correct 3D pore space geometry is of crucial importance for the determination of a realistic permeability. Recent advances have shown that machine learning image segmentation software can successfully be utilized to segment pore space in CT scans (Berg et al. 2018). The software *ilastik*, an interactive learning and segmentation toolkit by Berg et al. (2019), was used for the segmentation of the phases in our FIB-SEM images (Fig. 3). The built-in pixel classification module groups probabilities according to their different imaged features. In a manually controlled workflow, it was possible to reach high segmentation accuracies with only minor over- or under-estimations of the pore space (Fig. 3C, D).

### 3.5 Defining microporous domains

The need to define microporous domains results from the mismatch of permeability between µXCT simulations and gas-driven permeameter tests. While the simulation of permeability in structures with high permeability and porosity obtained by µXCT scans is precise, the effect of nanoporosity below resolution on permeability increases with decreasing permeability (Pittman and Thomas, 1979; Saxena et al. 2018). When comparing backscattered electron (BSE) images with µXCT images of the same slice, it becomes apparent that the smallest pores in µXCT images simplify the real pore structure (Figure A2). Furthermore, SEM and EDX images revealed that most of the pores are filled with clay minerals. Since both void and microporous regions share similar greyscale values, it is impossible to correctly differentiate upon segmentation. In this approach, the segmentation of the pores includes clay minerals with a low absorption contrast. To determine the distribution of illite in µXCT scans, we used correlative µXCT, SEM and EDX measurements. For this, a 3 mm plug was embedded into epoxy resin and then ground and polished until the region of interest was reached. Two sites were chosen for EDX mappings (Figure A2). Comparing EDX Mappings with CT images shows that the distribution of illite agrees with the textural findings of Fischer et al. (2012). Furthermore, it becomes apparent that illite enrichments coincide with regions that are usually referred to as void pore space in µXCT images. The mismatch between real pore structure and segmented pore space is highest in small pores and throats. To refine flow and reduce the influence of overestimated pore sizes in these specific regions, we define all pores with a diameter ≤ 2 voxels as microporous domains. The Brinkman term accounts small pores, where grains are porous themselves (Brinkman, 1949). To extract these regions from the initial pore space segmentation $F_{p*}$ we calculated Euclidian distance maps as used by Maurer et al. (2003):





$$d = \sqrt{(x_2 - x_1) + (y_2 - y_1) + (z_2 - z_1)} \qquad (6)$$

where $d$ is the distance between two points, $x_1, y_1$ and $z_1$ the coordinates of the first point, and $x_2, y_2$ and $z_2$ the coordinates

of the second point. $d(\leq 2)$ includes all pore voxels with the nearest distance of $\leq 2$ voxels to the next solid surface,

including the outer rim of larger pores. To remove this layer, a dilation is performed on $d(> 2)$ using a structure element

SE2 of $2 \times 2$ voxels. By subtracting the dilated image from $d(\leq 2)$, we get the microporous domain data $F_{\mu 2}$:

$$F_{\mu 2} = d(\leq 2) - \delta_{SE2}((d > 2)) \qquad (7)$$

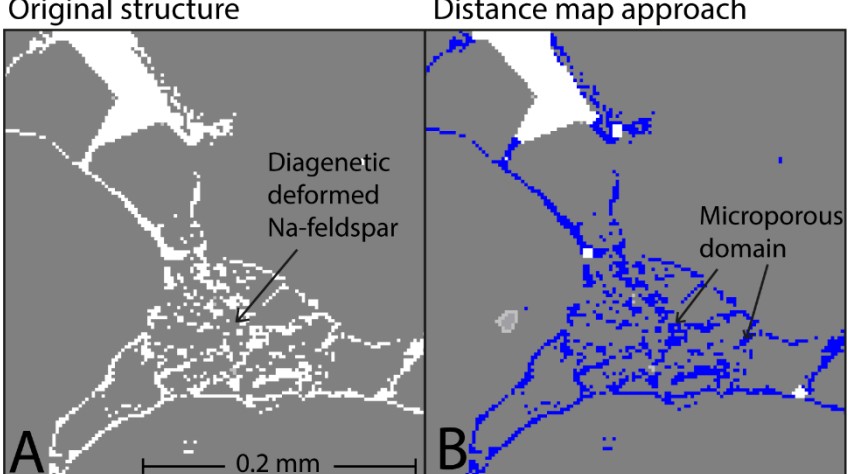

**Figure 4 (A) Segmented µXCT image of the pore space of a mini plug. (B) Image with applied microporous domain**
**using the Euclidian distance map approach**

In *GeoDict* the permeabilities of the microporous domains are calculated with the Brinkman term. Based on flow simulations

on 3D FIB-SEM images of illite meshworks, an isotropic permeability was assigned to the microporous domains (Fig. 4).

Yoon and Dewers (2013) confirmed the validity of the approach of extrapolating structural features of clays measured by

FIB-SEM to the pore scale. The complete workflow for a precise permeability simulation is illustrated in Fig. 5.





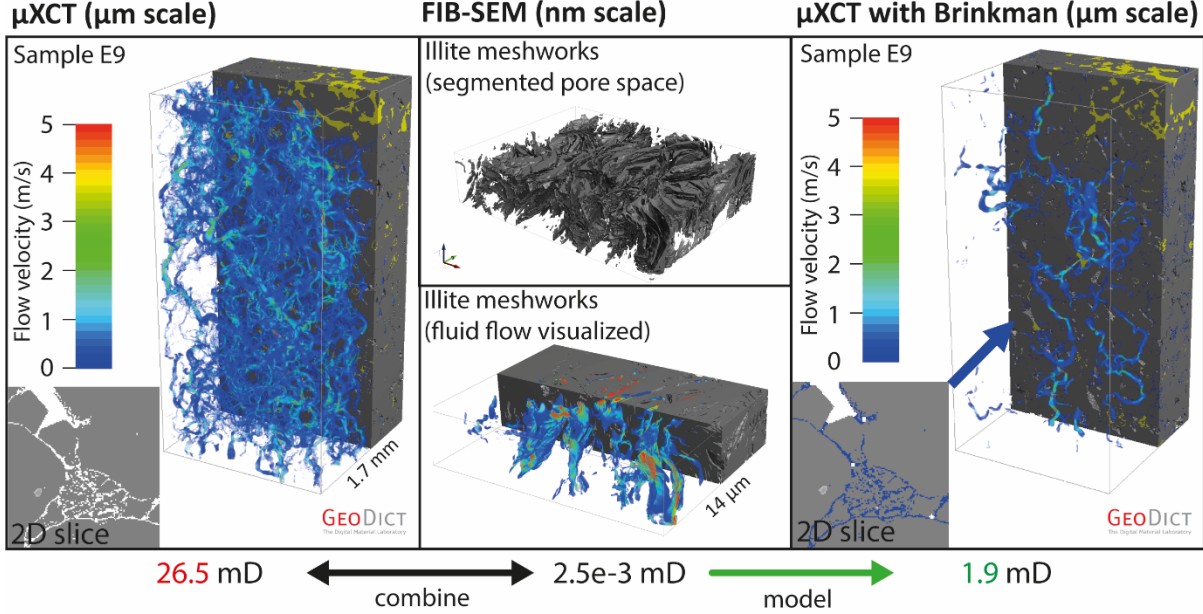

**Figure 5 Workflow for microporous domain modelling combining FIB-SEM nanoporosity with µXCT scans. Sample E9 of the aeolian layer with well-distributed flow paths was selected for visualisation. Upon the modelling of the microporous domains, the total fluid-flow velocity decreases. Displayed simulated permeabilities include the mean values of five samples with the best distribution of percolation paths. Fluid-flow velocities close to zero are transparent**

## 4 Results and discussion

### 4.1 Permeability measurements and pore size distribution

The measured intrinsic permeabilities range between 1.1 mD and 5.4 mD with an average of 2.9 mD for both aeolian facies (Table 1; Fig. 1, sample series E and F). The small variation of the physical rock properties between the main layers is induced by different angles of the grain layering in each plug sample and small-scale variations in grain and pore size distributions. The observed variance is in the typical range of observed permeability fluctuations for tight reservoir sandstones (Lis-Śledziona, 2019). Considering the observed permeability, the studied rock samples are at the lower end of the permeability range known for sedimentary reservoir rocks (Gluyas and Swarbrick, 2004).





**Table 1: Intrinsic permeabilities and porosities of the measured sandstone plug samples.**

|  | E5 | E6 | E7 | E8 | E13 | F4 | F6 | F8 | F9 | Mean | Standard deviation (±1σ) |
|---|---|---|---|---|---|---|---|---|---|---|---|
| **Permeability (mD)** | 1.1 | 5.1 | 5.4 | 2.2 | 4.2 | 2.3 | 2.3 | 1.9 | 1.4 | 2.9 | 1.5 |

The pore size distribution of the sub-sample analysed by mercury intrusion porosimetry (section 3.2) was obtained by a semi-logarithmic representation of the normalized intrusion volume achieved per pressure interval (Fig. 6). The consumed capillary stem intrusion volume of 27 % was within the acceptable range needed for precise measurements. We discovered a

major peak at approximately 1 µm, representing the most common pore throat diameter.

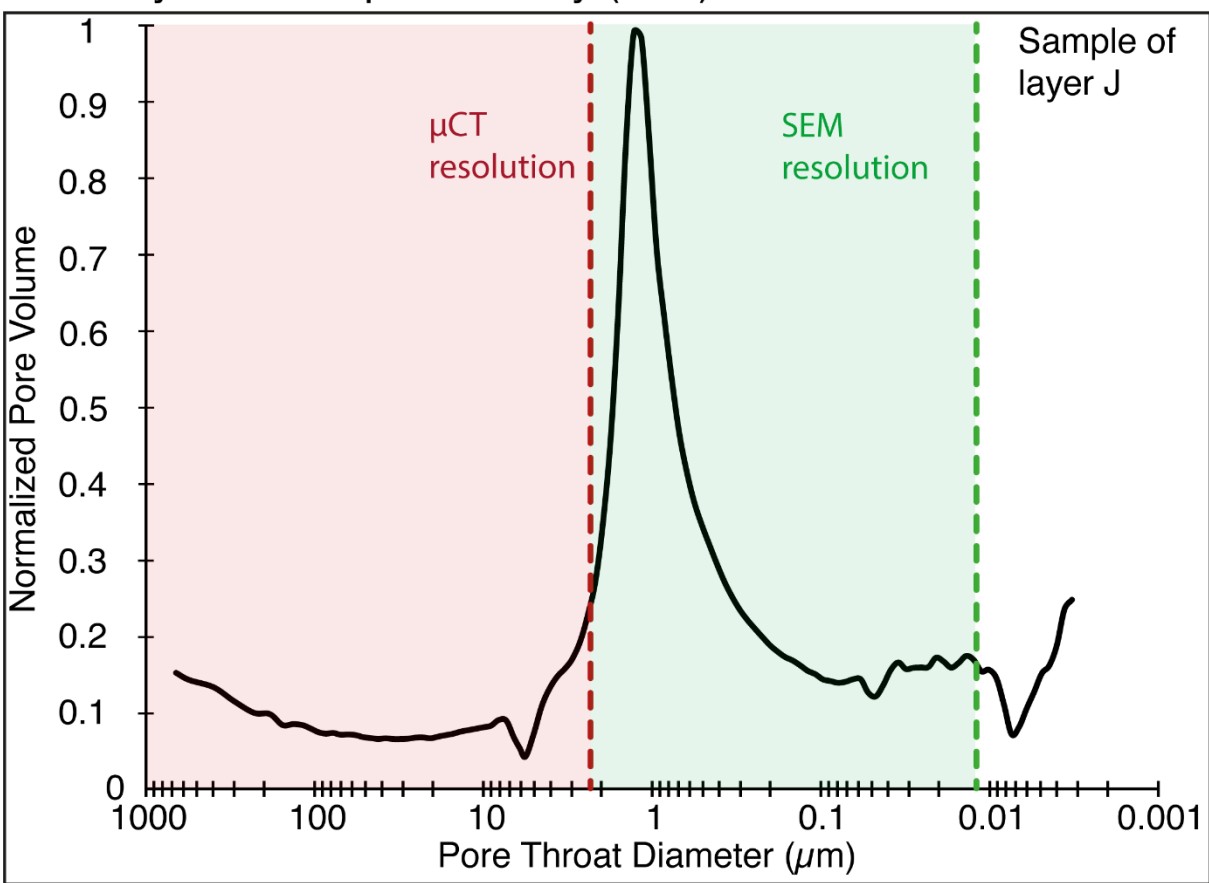

**Figure 6 Distribution of the pore throat diameters of a sample of the aeolian facies in layer J obtained from mercury intrusion porosimetry (MIP)**




In addition to the mercury porosimetry results, calculations of the pore size distribution of FIB-SEM and µXCT mini plug

images were conducted using the *GeoDict* module *PoroDict*. This module provides an algorithm that virtually pushes

spheres of different sizes into a medium to determine the 3D pore size distribution (Münch and Holzer, 2008). FIB-SEM

measurements of the pore size distribution cover the small pores related to the illite nanoporosity, while µXCT porosimetry

illustrates the pore size distribution of the larger scale intergranular pore space skeleton.

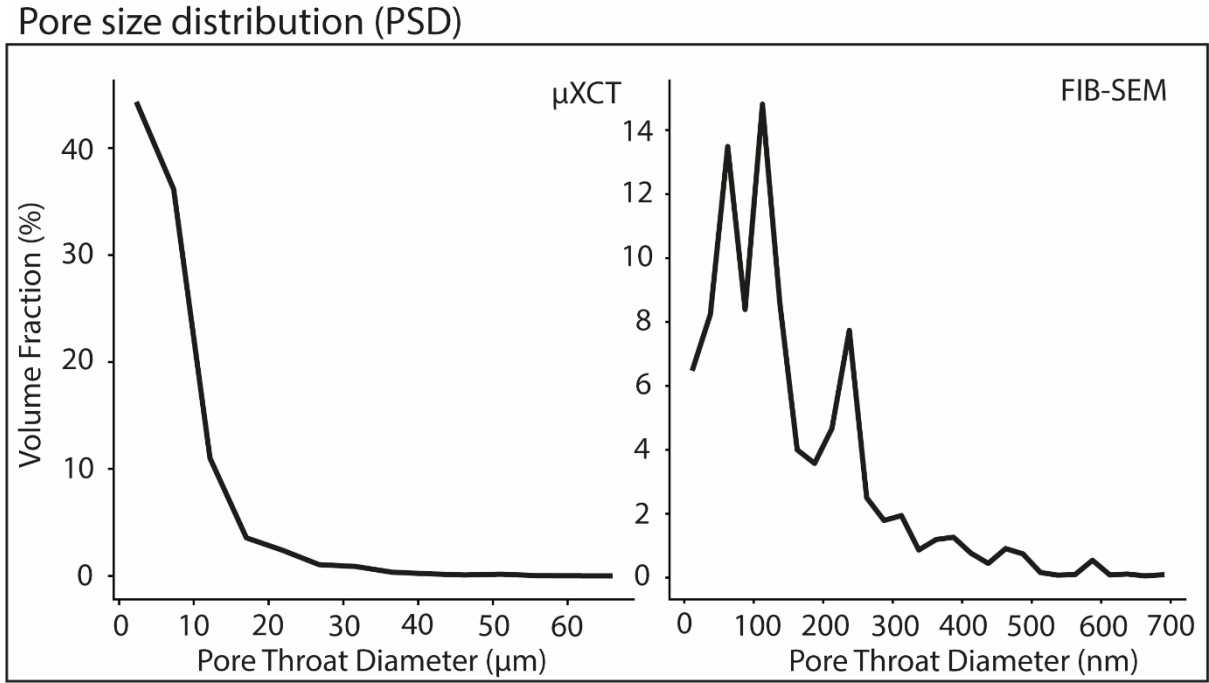

**Figure 7 Pore throat diameters from µXCT (left) and FIB-SEM images (right)**

The µXCT pore volume analysis shows a constant rise towards the smallest observed pore diameters (2.44 µm, Fig. 7). The

steepest slope is observed between 7.3 and 12.2 µm which indicates that the largest volume of pore bodies is observed in that

range. The calculated pore size distribution from nanoporous 3D structures of illite meshworks obtained by FIB-SEM

imaging shows three distinct peaks in its at 75 nm, 125 nm and 250 nm. This illustrates that the most common inscribed pore

diameters are relatively small compared to the actual extent of the pores, which is considered typical for slit-shaped pore

systems where the pore axial ratio is high (Desbois et al. 2016). Ultimately, this results from the different thicknesses of clay

platelets and heterogeneity in the alignment of the illite nanostructure (Aplin et al. 2006).





An apparent gap exists between diameters observed by MIP and 3D imaging with FIB-SEM and µXCT. While MIP peaks at around 1 µm, diameters observed by µXCT start at 2.4 µm and the largest inscribed diameters observed by FIB-SEM are

below 700 nm. However, it must be noted that pressure-controlled MIP generally gives information about the number of pore throats whereas pore size distributions provided by imaging techniques give information about pore body volumes (Zhao et al. 2015). Furthermore, pore shielding may cause an underestimation of larger pores for MIP (Abell et al. 1999; Gane et al. 2004). The occurrence of authigenic illites is the likely cause of this effect as they are commonly found in pore throat areas. Since the Washburn equation assumes ideal pore throats of cylindrical shape, the underestimation of larger

pores becomes more evident with the increasing complexity of the pore throat system at both the mm and nm scale (Washburn, 1921). Therefore, a direct comparison between the used methods is unlikely to result in compatible results.

## 4.2 Permeability simulation

A calculated isotropic permeability of $2.5 \cdot 10^{-18} \, m^2$ was used for the microporous domains based on the Navier-Stokes fluid-flow simulations of permeability of FIB-SEM scans of the illite meshworks. The number of porous voxels resulting

from clay mineral modelling ranges between 3.3 vol.-% and 7.1 vol.-% of the total structure volume for all considered µXCT scans. A comparison of the modelled clay minerals in µXCT scans with XRD mass balancing highlights a large difference between the measured mineral abundances (Table 2, EDX analysis).

**Table 2: Modelled clay mineral content within the microporous domains in µXCT scans**

|  | C3 | D2 | D9 | E9 | F7 | G2 | G8 | H9 | I3 | I9 | J3 | J7 | Mean | Standard deviation (±1σ) |
|---|---|---|---|---|---|---|---|---|---|---|---|---|---|---|
| **Clay mineral content (vol.-%)** | 4.7 | 3.3 | 5.0 | 5.4 | 7.1 | 5.0 | 6.0 | 6.3 | 4.1 | 6.2 | 6.2 | 6.1 | 5.4 | 1.0 |

While the mean amount of clay minerals based on XRD measurements was 12.7 wt.-% (about 11.3 vol.-% within a structure with 8 % porosity), an average amount of 5.4 vol.-% was modelled by the distance map algorithm. This is expected since the illite content inside grains was not modelled since it has no effect on permeability. We simulated permeability of 12 mini plug samples that were scanned by µXCT and compared them with measurements from gas permeameter experiments. As a





first step, we extracted and illustrated the ten and hundred largest open flow paths through the pore space of all mini plug

cores before the modelling of the microporous domains (Fig. 8). This yields information concerning the heterogeneity of the

flow fields and allows to check the validity of the Navier-Stokes simulations. Structures with flow impingement often cause

numerical problems which results in an artificial underestimation of the permeability simulations. Significant

underestimations of permeability after clay mineral modelling were also found in areas where the percolation paths in the

samples were limited to a few voxels in the structure before modelling. This effect leads to an artificial permeability drop,

which renders calculations to be less precise. Since evenly distributed flow paths are necessary to determine the true

permeability of a volume of a rock, we considered only permeability calculations of samples which show no flow

impingement for modelling (Bear, 1972; Leu et al. 2014; Zhang et al. 2000).

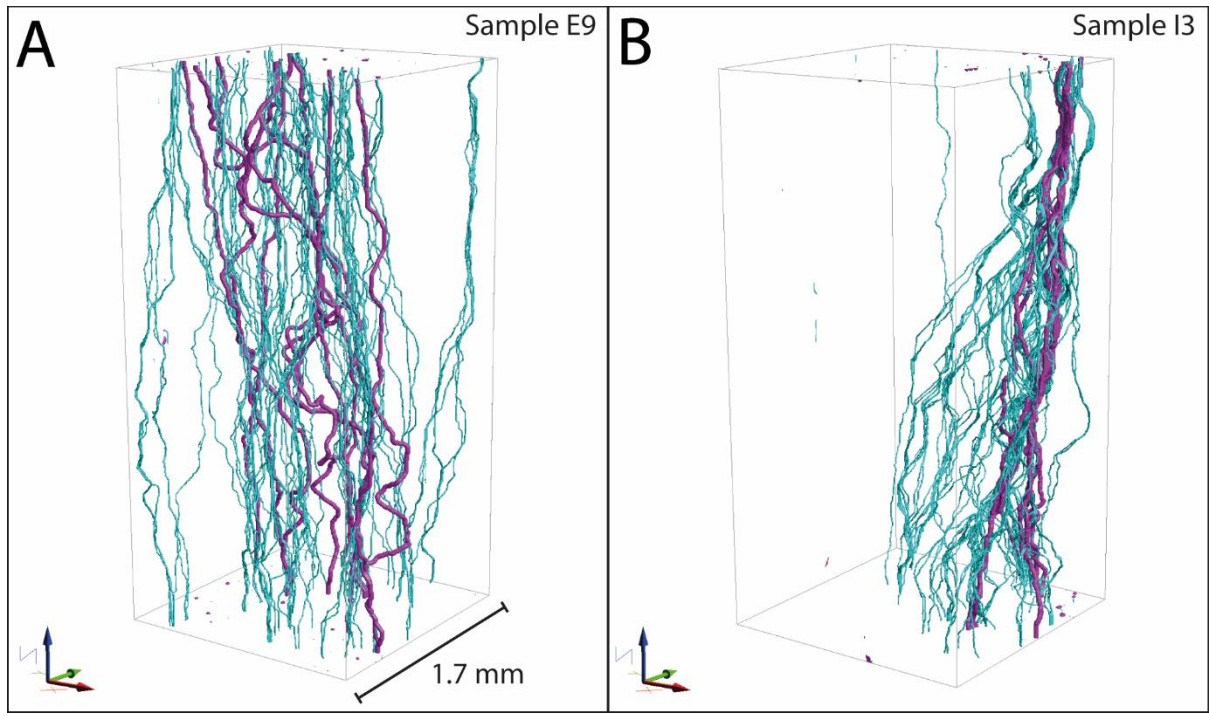

**Figure 8 Comparison of percolation paths in two µXCT reconstructions of the 3D imaging data without microporous**
**domain modelling (A) Well distributed percolation paths in a µXCT reconstruction of mini plug sample E9 without**
**microporous domain modelling. (B) Constricted percolation paths with flow impingement limited to a small region of**
**the structure in mini plug sample I3 without microporous domain modelling. The ten largest percolation paths are**
**coloured in purple, the hundred largest percolation paths are coloured in cyan.**






The mean value of the experimentally obtained intrinsic gas permeabilities was 2.9 mD (Table 1, Fig. 9A). Samples with no

flow impingement had an initial mean permeability of 26.5 mD, whereas the mean permeability simulated on µXCT images

was 26.5 (Fig. 9B). With applied microporous domains, a mean permeability of 1.9 mD was calculated (Fig. 9C).

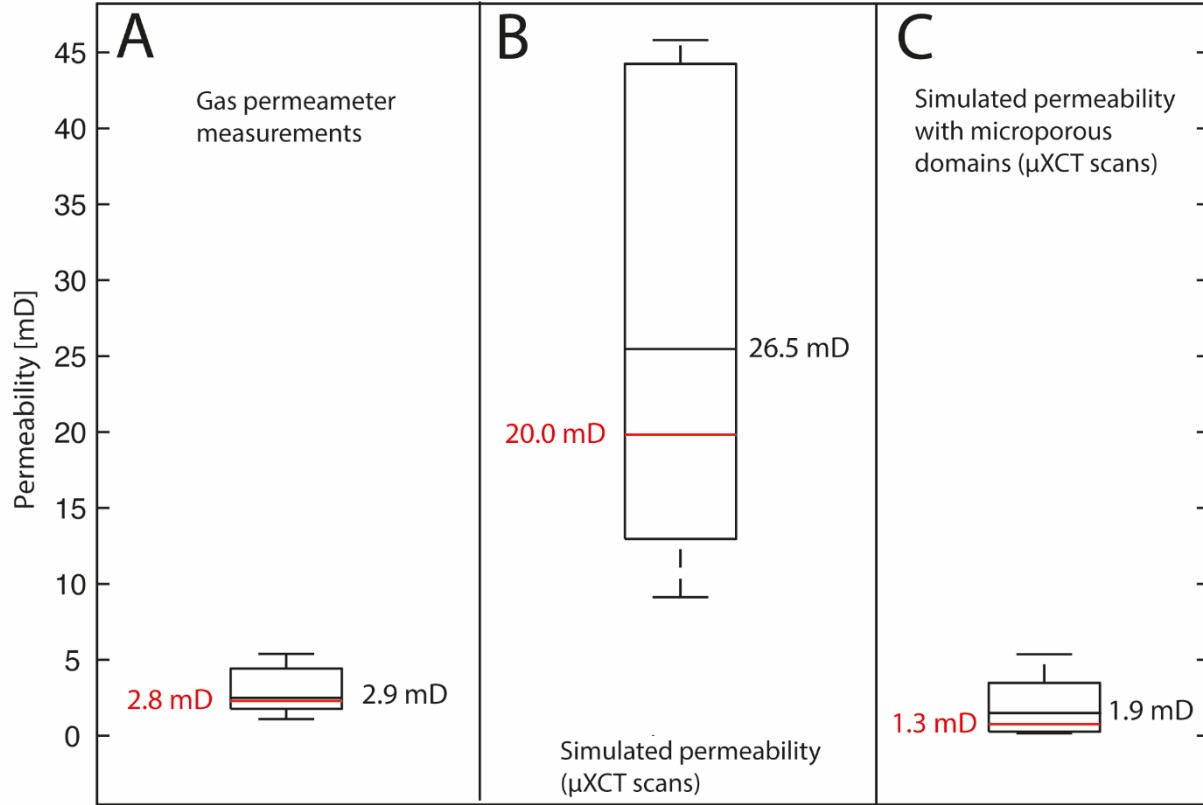

**290** **Figure 9 Comparison of measured and simulated permeabilities. The boxes indicate the upper and lower quartile of derived permeability values. Median values are coloured in red, mean values are coloured in black. (A) Measured permeabilities using the gas permeameter test. (B) Simulated permeabilities using µXCT data. (C) Simulated permeabilities using µXCT data with modelled microporous domains.**

The mismatch between measured and simulated permeabilities could be decreased to 1 mD (-34.5 %) compared to 23.6 mD

**295** (+813.8 %) prior to the modelling. Hence, our approach significantly improved the match between measured and simulated

permeabilities and lies within the standard deviation of the permeabilities measured by the gas permeameter (Table 1).

Histograms of the simulated fluid-flow velocities indicate a strong decrease in velocities due to the modelling of the

microporous domains in sample E9 (Fig. 10). Correlation histograms of the velocities depict differences induced by the

modelling. While identical fluid-flow field velocities would plot as a single straight line, the velocities, in this case, show



straight lines with varying slopes. This indicates a general decrease of the fluid-flow velocities with a splitting of different

flow path velocities. Points which plot as a line represent a main fluid-flow path with a direct correlation of the velocities

between the original and the modelled structures flow field. Furthermore, a wider spread of the distribution of high velocities

results from microporous domain modelling. This indicates narrow pore throats where fluid-flow velocities are locally

enhanced compared to the structure before the modelling.

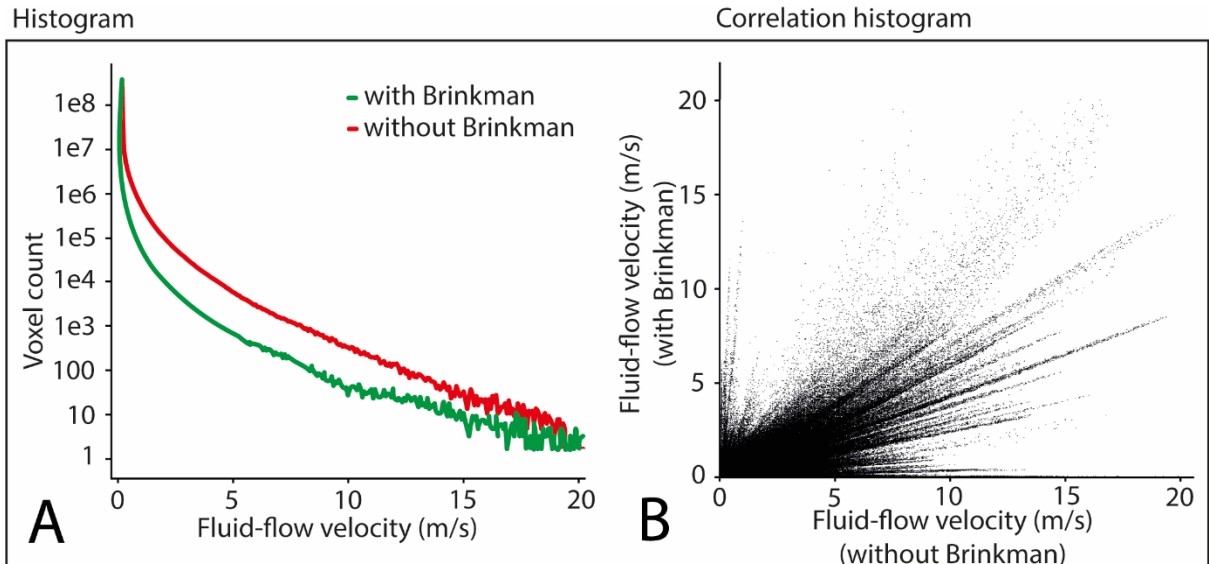


**Figure 10 Comparison of unmodelled µXCT images and modelled microporous domain images (A). Correlation histogram prior to and after the modelling that shows a decrease of the main fluid-flow velocities, while few velocities after the modelling were increased with a wide spread of distribution (B).**

Based on our combined analytical and numerical study, further research may help to increase the accuracy of simulated

permeabilities even further. Since isotropic permeabilities of the microporous domains were applied to µXCT images, the

accuracy can be improved by taking the anisotropy of clay mineral fabrics and surface topology into account. This can be

done by applying anisotropic permeabilities in the calculation of the microporous domains. While this study showed a good

match between the experimental and simulated permeability, the need to include heterogeneities of clay mineral layering to

improve the simulations was depict in Villiéras et al. (1997).



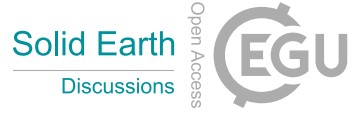

## 5 Conclusions

Overall, the outcome of this study shows that combining µXCT and FIB-SEM imaging with numerical models constitutes a valuable and novel approach for determining physical properties of clay-bearing tight reservoir rocks. Considering the high number of accessible pores in the scans, the phenomenon of flow impingement was mainly attributed to the unresolvable nanoporosity. While permeability, which is one of the most important reservoir properties, is often determined by simulations based on µXCT scans of small samples taken from a field-scale reservoir, we could demonstrate that an accurate estimation for clay-rich and low permeable rocks is only possible if nanoscale porosity is also included. Thus, our simulations using the Euclidian distance map approach resulted in an improved match with stationary gas permeameter measurements in contrast to permeability simulations merely based on unmodified µXCT images. Adopting this multi-method approach, we increased the accuracy of simulated permeabilities of samples measured by µXCT. These results have important implications for improved modelling of reservoirs relevant to gas and water applications. A realistic simulated permeability of a tight reservoir sandstone could only be achieved by appropriate modelling of the nanoporosity related to matrix clay minerals (illite) that occur below the µXCT resolution. The simulated permeability based on combined µXCT and FIB-SEM images and modelled microporous domains showed good agreement with the experimental results. Obtaining an even distribution of the simulated fluid-flow paths through the sample without flow impingement was necessary to obtain an accurate permeability estimation from 3D imaging. Resolving the nanopore structure and distribution of clay mineral-related features by the combined analytical and numerical modelling approach represents a tool for achieving a more accurate understanding of the fluid flow behaviour within tight sandstones, with direct relevance to predicting the injection, storage or extraction of gas or water in a reservoir rock. Our multi-method approach can be applied to determine more accurate permeability values and flow paths for reservoir rocks with high clay mineral contents if direct experimental measurements are not successful. Hence, future studies should focus on distinguishing the different morphologies of clay minerals and their related anisotropic effect on rock permeability. While the permeability of the nanoporous structures depends highly on the layering of the clays and their spatial orientation on the grain surfaces and within feldspars. This approach should include a variety of various sedimentological facies also with high porosity and permeability investigating whether clay mineral modelling is also a valid tool for such sedimentary rocks.





**Appendices**

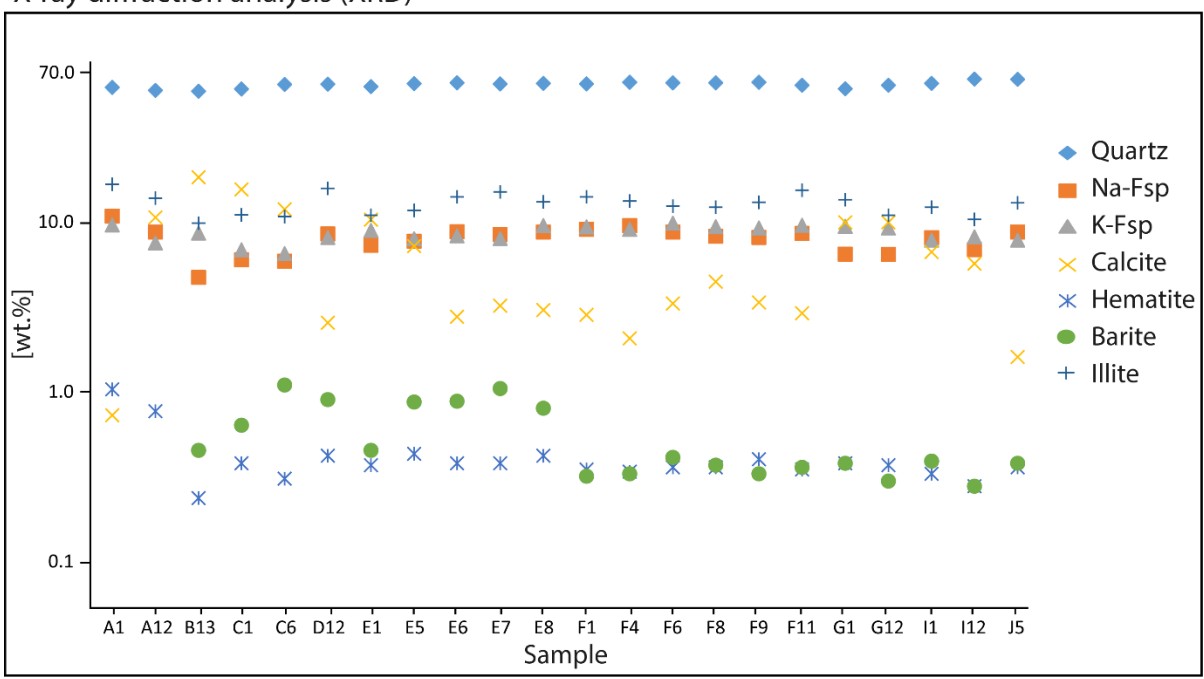

**Figure A1 XRD analysis of the main minerals in the plugs based on Rietveld calculations, determined for several samples of facies A to J of the sandstone block.**





**Figure A2 (A) µXCT and corresponding (B) BSE image of the polished section including the close-up regions S1 and S2 that were used for EDX mapping to analyse the distribution of illites. (C) Ternary plot of Ca, K and Al content of each pixel in the mappings. Calcite, illite and K-feldspar plot in 3 distinct regions. By selecting these areas, a phase map overlay was created that visualises the location of these phases in SE images. (D) Region S1 and S2 with phase map overlays. Illite is predominantly found in the vicinity of altered feldspars, as coating along grain boundaries and as authigenic pore filling.**



**Author contribution**

Conceptualization, A.J., M.P., S.H., F.E., L.N.W., G.G., P.B. and M.K., Methodology, A.J. and M.P., Software, A.J., M.P.,
F.E., Validation, A.J., M.P., S.H., F.E., L.N.W., G.G., P.B. and M.K., Formal analysis, A.J. and M.P., Investigation, A.J.,
M.P., S.H., and O.M., Resources, F.E., L.N.W., G.G., P.B. and M.K., Data curation, A.J., M.P., S.H., F.E., O.M., L.N.W.,
G.G., P.B., M.K., Writing – Original Draft, A.J., Writing – Review & Editing, A.J., M.P., S.H., F.E., O.M., L.N.W., G.G.,
P.B., M.K., Visualization, A.J. and M.P., Supervision, F.E., L.N.W., G.G., P.B. and M.K., Project Administration, F.E.,
L.N.W., G.G., P.B. and M.K., Funding Acquisition, F.E., L.N.W., G.G., P.B. and M.K.


**Competing interests**

The authors declare that they have no conflict of interest.

**Acknowledgements**

This work was supported by the German Federal Ministry of Education and Research (BMBF) "Geological Research for
Sustainability (GEO:N)" program, which is part of the BMBF "Research for Sustainable Development (FONA3)"
framework program. It is part of the project ResKin (Reaction kinetics in reservoir rocks, 03G0871E). We would like to
thank Fabian Wilde and the staff of PETRA synchrotron facility at DESY Hamburg for their assistance at the imaging
beamline P05. Jens Hornung and Meike Hintze from the TU Darmstadt are also acknowledged for enabling us gas
permeability measurements of the studied samples.

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
