# Peer review of "Simulating permeability reduction by clay mineral nanopores in a tight sandstone by combining $\mu XCT$ and FIB-SEM imaging"

_Solid Earth, 2020_

## Short Comment (SC1) · 7 Oct 2020

The manuscript presents sandstone samples' imaging at different resolutions to account for the role of nano clays in the permeability of the porous media. They implemented the Navier-Stokes-Brinkman simulation technique to model the flow behavior of the sandstone samples containing clays. The paper is well-written and of great interest to the technical readers. Different imaging techniques, including micro-CT, SEM, FIB-SEM, and EDX, were successfully implemented to characterize the sandstone samples. The imaging techniques were properly described and the techniques for laboratory measurements of the real rock sample. The data support conclusions.

[Figure]

However, I believe that the author should address the following points too:

The paper's Appendix comparing micro-CT, BSE, and EDX should be merged into the main text. FigA provides a clear insight into the "problem definition" of the paper. The readers should wait until reading this Appendix and see this figure to understand why Illites are not visible in micro-CT images.

-Authors considered a 2-voxel pore layer next to solid surfaces as pore locations for illites in the primary simulation model. The reason for selecting this value was not discussed in the paper. Also, the methodology used in Eq. 7 requires more detailed explanations.

The authors well-explained the impact of adding porous voxels to the micro-CT image and discussed its impact on the domain's permeability value. However, the discussion of the same effect on the porosity of the system is missing.

---

## Author Comment (AC1) · 20 Oct 2020

**Authors comment**

**Simulating permeability reduction by clay mineral nanopores in a tight sandstone by combining µXCT and FIB-SEM imaging**

Arne Jacob[1], Markus Peltz[2], Sina Hale[3], Frieder Enzmann[1], Olga Moravcova[1], Laurence N. Warr[2], Georg Grathoff[2], Philipp Blum[3] and Michael Kersten[1]

*Please note: All our responses to remarks of reviewers are in red and italic.*

SC:

The paper's Appendix comparing micro-CT, BSE, and EDX should be merged into the main text. FigA provides a clear insight into the "problem definition" of the paper. The readers should wait until reading this Appendix and see this figure to understand why Illites are not visible in micro-CT images.

*AC:*

*As suggested by Mr. Sadeghnejad, we inserted the images into the manuscript text section. This improves the comprehensibility of the manuscript, clarifies the motivation, and illustrates why we had to model illites into µXCT structures to obtain the true rock permeability.*

SC:

Authors considered a 2-voxel pore layer next to solid surfaces as pore locations for illites in the primary simulation model. The reason for selecting this value was not discussed in the paper. Also, the methodology used in Eq. 7 requires more detailed explanations.

*AC:*

*We updated the description of the decision why we chose a 2-voxel layer of illites and included it into the text. We updated the description of the method in line 193-194. Furthermore, we updated the text from line 207-211. We changed fig. 4 and the figure description to improve the comprehensibility of the method.*

SC:

However, the discussion of the same effect on the porosity of the system is missing.

*AC:*

*With a focus on the permeability impact of clay minerals, this work shows that only a minor alteration of the pore throat morphology can be accompanied by a strong reduction of permeability in tight reservoir rocks. Assuming porous voxels, the porosity does not change significantly by applying the methodology. It is our intention to emphasize that the well-known relationship between porosity and*

40    permeability changes with a disconnection of the pore throats (e.g. Leu et al., 2014). The impact on permeability of this effect increases with decreasing permeability of the considered rock.

45  Appendix:

Leu, L., Berg, S., Enzmann, F. *et al.* Fast X-ray Micro-Tomography of Multiphase Flow in Berea Sandstone: A Sensitivity Study on Image Processing. *Transp Porous Med* **105,** 451–469 (2014). https://doi.org/10.1007/s11242-014-0378-4

---

## Referee Comment (RC1) · Anonymous Referee #1 · 2 Nov 2020

This paper investigates the effect of clay mineral in nanopores on the overall permeability in porous rocks. The main innovation is the combination of imaging method with machine learning and Brinkman solutions. Very few studies have done this before. This paper is well-written with explicit structure, however, there are some comments needed to be addressed: 1. Line 40-41, in the Introduction section, the following references (Liu and Mostaghimi 2018; Liu et al. 2020) can't be ignored to discuss the recent advances in the pore-scale studies below $\mu$XCT resolution. Liu, M., & Mostaghimi, P. (2018). Reactive transport modelling in dual porosity media. Chemical Engineering Science, 190, 436-442. Liu, M., Starchenko, V., Anovitz, L. M., & Stack, A. G. (2020). Grain detachment and transport clogging during mineral dissolution in carbonate rocks with

permeable grain boundaries. Geochimica et Cosmochimica Acta, 280, 202-220. 2. Line 87, "The term . . ..", this sentence seems have different font style with the context. Please check it through the paper 3. Line 89, reference is needed for the definition of "nanoporous" 4. Line 208, can the authors provide the values of the parameters used in Equation (1)? 5. Scale bar is missing for Figure 2. 6. Line 175, I think there should be more introduction about the machine learning method used in Figure 3.

---

## Referee Comment (RC2) · Anonymous Referee #2 · 7 Nov 2020

This paper demonstrates a new method to calculate the permeability of the tight sandstone with clay mineral nanopores by simulation. This is a very interesting and important topic for the development of pore scale imaging method. The authors combined micro-CT images for macro pores and FIBSEM images for clay mineral nanopores to capture all the pores and throats. Then Geodict was used for simulation. With the consideration to the suggested comments, as well as other reviewers' comments, this paper is suggest to for publication.

Here are some comments:

1. Line 104, list the brand, type and accuracy of the differential pressure transducer.

2. Line 111, MIP should be replaced by MICP. 3. Line 128, add more details of the machine learning segmentation used in this study. 4. Line 156, how did the authors calculate the error bound? 5. Line 258, modify the absolute permeability format. 6. The unit of the permeability should be consistent for the whole paper. 7. Additional question: is this method applicable for shale?

---

## Author Comment (AC2) · 13 Nov 2020

**Authors comment**

**Simulating permeability reduction by clay mineral nanopores in a tight sandstone by combining µXCT and FIB-SEM imaging**

Arne Jacob[1], Markus Peltz[2], Sina Hale[3], Frieder Enzmann[1], Olga Moravcova[1], Laurence N. Warr[2], Georg Grathoff[2], Philipp Blum[3] and Michael Kersten[1]

The authors thank the referees for their time spent reviewing our manuscript. We appreciate the referees' comments and suggestions and agree with all the referees' statements. The corrections and suggested changes have been incorporated in the revised manuscript. The modified or added sentences can easily be found in the version where line numbers of the modifications are indicated in our specific responses provided in the table below.

|  | Anonymous referee #1: | Authors' reply: |
|---|---|---|
| 1. | Line 40-41, in the Introduction section, the following references (Liuand Mostaghimi 2018; Liu et al. 2020) can't be ignored to discuss the recent advances in the pore-scale studies below µXCT resolution. Liu, M., & Mostaghimi, P. (2018). Reactive transport modelling in dual porosity media. Chemical Engineering Science,190, 436-442. Liu, M., Starchenko, V., Anovitz, L. M., & Stack, A. G. (2020). Grain detachment and transport clogging during mineral dissolution in carbonate rocks with C1SED Interactive comment Printer-friendly version Discussion paper permeable grain boundaries. Geochimica et Cosmochimica Acta, 280, 202-220. | We appreciate your comment and added the suggested references to line 40-42. |
| 2. | Line 87, "The term....", this sentence seems have different font style with the context. Please check it through the paper. | We checked for different font styles and corrected it throughout the manuscript. |

| | | |
|---|---|---|
| 3. | Line 89, reference is needed for the definition of "nanoporous". | We added a description for the term "nanoporous" from line 88-90 (e.g. Tinet et al. 2020). |
| 4. | Line 208, can the authors provide the values of the parameters used in Equation (1)? | Since line 208 does not contain information about the equation, we think you might have meant line 108. We added a section about the physical parameters which were used for the gas permeameter test (line 244-254). These include values of the applied differential pressure and size of the samples. Furthermore, we added an example for the applied pressures for sample F8 (Table 1). |
| 5. | Scale bar is missing for Figure 2. | In Figure 3 (former Figure 2), there is a scale bar in the bottom left corner of the figure. |
| 6. | Line 175, I think there should be more introduction about the machine learning method used in Figure 3. | *Ilastik* uses a Random Forest algorithm to provide different trees for the classification of the voxels/pixels. This is used to achieve a segmentation based on not only greyscale values, but shape as well. We added a section about the machine learning method from line 185-191. |

15

**References:**

Tinet, A.-J., Corlay, Q., Collon, P., Golfier, F., Kalo, K.: Comparison of various 3D pore space reconstruction methods and implications on transport properties of nanoporous rocks, Advances in
20  Water Resources, 141, doi: 10.1016/j.advwatres.2020.103615, 2020.

---

## Author Comment (AC3) · 13 Nov 2020

**Authors comment**

**Simulating permeability reduction by clay mineral nanopores in a tight sandstone by combining µXCT and FIB-SEM imaging**

5  Arne Jacob[1], Markus Peltz[2], Sina Hale[3], Frieder Enzmann[1], Olga Moravcova[1], Laurence N. Warr[2], Georg Grathoff[2], Philipp Blum[3] and Michael Kersten[1]

The authors thank the referees for their time spent reviewing our manuscript. We appreciate the

referees' comments and suggestions and agree with all the referees' statements. The corrections and

10  suggested changes have been incorporated in the revised manuscript. The modified or added sentences

can easily be found in the version where line numbers of the modifications are indicated in our specific

responses provided in the table below.

|  | Anonymous referee #2: | Authors' reply: |
|---|---|---|
| 1. | Line 104, list the brand, type and accuracy of the differential pressure transducer. | We listed the brand, type and accuracy of the equipment we have used from line 108-111. |
| 2. | Line 111, MIP should be replaced by MICP. | We replaced the term "MIP" by "MICP" throughout the text. |
| 3. | Line 128, add more details of the machine learning segmentation used in this study | Following the referees' suggestion, we added a section about the machine learning (line 185-191). We explain how the algorithm classifies pixel/voxel using the random forest classifier. |
| 4. | Line 156, how did the authors calculate the error bound? | The error bound is one of the mandatory settings for the solver. Since every calculated permeability is a result of an iterative differential equation process, the "true" value for permeability can only be approximated. A low error bound value of 0.05 often requires a simulation time of days to reach the specified stopping criterion. We added a description in the manuscript from line 164-167. |
| 5. | Line 258, modify the absolute | We changed the format of the permeability to mD. |

| | | permeability format. | |
|---|---|---|---|
| 6. | The unit of the permeability should be consistent for the whole paper | We made sure that the permeability format is consistent in the manuscript. |
| 7. | Additional question: Is this method applicable for shale? | Thank you for asking. Since the structure of shales is very different compared to sandstones, the method might be applicable depending on the pore throat characteristics. In shales, the dominating pore structure can be either fractured or porous while organic matter may play a role too (Tiwari et al. 2013, Grathoff et al. 2016). That is why an investigation of the method using mudrocks might be the most natural next step to climb. In the porous case, the method might be applicable when the pores are large enough while in the other cases the modelled clay mineral content might have no or only a minor effect on the permeability calculations. Further studies are necessary to gain knowledge about the effect of clay modelling on permeability in shales when our method should be applied. Of course, this is a very promising topic, and the method should be benchmarked on these structures as well. We added two sentences about the possible applicability in other types of rocks from line 351-354. |

15

**References:**

P. Tiwari, M. Deo, C.L. Lin, J.D. Miller,
Characterization of oil shale pore structure before and after pyrolysis by using X-ray micro CT, Fuel,
20  Volume 107, doi: 10.1016/j.fuel.2013.01.006, 2013

Grathoff, G. H., Peltz, M., Enzmann, F., and Kaufhold, S.: Porosity and permeability determination of organic-rich Posidonia shales based on 3-D analyses by FIB-SEM microscopy, Solid Earth, 7, 1145–1156, https://doi.org/10.5194/se-7-1145-2016, 2016.

---

## Author Comment (AC4) · 13 Nov 2020

**Authors comment**

**Simulating permeability reduction by clay mineral nanopores in a tight sandstone by combining µXCT and FIB-SEM imaging**

5  Arne Jacob[1], Markus Peltz[2], Sina Hale[3], Frieder Enzmann[1], Olga Moravcova[1], Laurence N. Warr[2], Georg Grathoff[2], Philipp Blum[3] and Michael Kersten[1]

|  |  | Saeid Sadeghnejad: | Authors' reply: |
|---|---|---|---|
| 1. |  | The paper's Appendix comparing micro-CT, BSE, and EDX should be merged into the main text. Fig A provides a clear insight into the "problem definition" of the paper. The readers should wait until reading this Appendix and see this figure to understand why Illites are not visible in micro-CT images. | As suggested by Mr. Sadeghnejad, we inserted the images into the manuscript text section (Figure 2,5). This improves the comprehensibility of the manuscript, clarifies the motivation, and illustrates why we had to model illites into µXCT structures to obtain the true rock permeability. |
| 2. |  | Authors considered a 2-voxel pore layer next to solid surfaces as pore locations for illites in the primary simulation model. The reason for selecting this value was not discussed in the paper. Also, the methodology used in Eq. 7 requires more detailed explanations. | We updated the description of the decision why we chose a 2-voxel layer of illites and included it into the text. We updated the description of the method in line 213-216. Furthermore, we updated the text from line 221-224. We changed fig. 6 and the figure description to improve the comprehensibility of the method. |
| 3. |  | However, the discussion of the same effect on the porosity of the system is missing. | With a focus on the permeability impact of clay minerals, this work shows that only a minor alteration of the pore throat morphology can be accompanied by a strong reduction of permeability in tight reservoir rocks. Assuming porous voxels, the porosity does not change significantly by applying the methodology. It is our intention to emphasize that the well-known relationship between porosity and permeability changes with a disconnection of the pore throats (e.g. Leu et al., 2014). The impact on permeability of this |

| | | effect increases with decreasing permeability of the considered rock. |
| --- | --- | --- |

10  Appendix:

Leu, L., Berg, S., Enzmann, F. *et al.* Fast X-ray Micro-Tomography of Multiphase Flow in Berea Sandstone: A Sensitivity Study on Image Processing. *Transp Porous Med* **105,** 451–469 (2014). https://doi.org/10.1007/s11242-014-0378-4

---

## Editor Decision (ED1)

| | Saeid Sadeghnejad: | Authors' reply: |
|---|---|---|
| 1. | The paper's Appendix comparing micro-CT, BSE, and EDX should be merged into the main text. Fig A provides a clear insight into the "problem definition" of the paper. The readers should wait until reading this Appendix and see this figure to understand why Illites are not visible in micro-CT images. | As suggested by Mr. Sadeghnejad, we inserted the images into the manuscript text section (Figure 2,5). This improves the comprehensibility of the manuscript, clarifies the motivation, and illustrates why we had to model illites into µXCT structures to obtain the true rock permeability. |
| 2. | Authors considered a 2-voxel pore layer next to solid surfaces as pore locations for illites in the primary simulation model. The reason for selecting this value was not discussed in the paper. Also, the methodology used in Eq. 7 requires more detailed explanations. | We updated the description of the decision why we chose a 2-voxel layer of illites and included it into the text. We updated the description of the method in line 213-216. Furthermore, we updated the text from line 221-224. We changed fig. 6 and the figure description to improve the comprehensibility of the method. |
| 3. | However, the discussion of the same effect on the porosity of the system is missing. | With a focus on the permeability impact of clay minerals, this work shows that only a minor alteration of the pore throat morphology can be accompanied by a strong reduction of permeability in tight reservoir rocks. Assuming porous voxels, the porosity does not change significantly by applying the methodology. It is our intention to emphasize that the well-known relationship between porosity and permeability changes with a disconnection of the pore throats (e.g. Leu et al., 2014). The impact on permeability of this effect increases with decreasing permeability of the considered rock. |
| | Anonymous referee #1: | |
| 1. | Line 40-41, in the Introduction section, the following references (Liuand Mostaghimi 2018; Liu et al. 2020) can't be ignored to discuss the recent advances in the pore-scale studies below µXCT resolution. Liu, M., & Mostaghimi, P. (2018). Reactive transport modelling in dual porosity media. Chemical Engineering Science,190, 436-442. Liu, M., Starchenko, V., Anovitz, L. M., & Stack, A. G. (2020). Grain detachment and transport clogging during mineral dissolution in carbonate rocks with | We appreciate your comment and added the suggested references to line 40-42. |

| | | |
|---|---|---|
| | C1SED Interactive comment Printer-friendly version Discussion paper permeable grain boundaries. Geochimica et Cosmochimica Acta, 280, 202-220. | |
| 2. | Line 87, "The term....", this sentence seems have different font style with the context. Please check it through the paper. | We checked for different font styles and corrected it throughout the manuscript. |
| 3. | Line 89, reference is needed for the definition of "nanoporous". | We added a description for the term "nanoporous" from line 88-90 (e.g. Tinet et al. 2020). |
| 4. | Line 208, can the authors provide the values of the parameters used in Equation (1)? | Since line 208 does not contain information about the equation, we think you might have meant line 108. We added a section about the physical parameters which were used for the gas permeameter test (line 244-254). These include values of the applied differential pressure and size of the samples. Furthermore, we added an example for the applied pressures for sample F8 (Table 1). |
| 5. | Scale bar is missing for Figure 2. | In Figure 3 (former Figure 2), there is a scale bar in the bottom left corner of the figure. |
| 6. | Line 175, I think there should be more introduction about the machine learning method used in Figure 3. | *Ilastik* uses a Random Forest algorithm to provide different trees for the classification of the voxels/pixels. This is used to achieve a segmentation based on not only greyscale values, but shape as well. We added a section about the machine learning method from line 185-191. |
| | Anonymous referee #2: | |
| 1. | Line 104, list the brand, type and accuracy of the differential pressure transducer. | We listed the brand, type and accuracy of the equipment we have used from line 108-111. |
| 2. | Line 111, MIP should be replaced by MICP. | We replaced the term "MIP" by "MICP" throughout the text. |
| 3. | Line 128, add more details of the machine learning segmentation used in this study | Following the referees' suggestion, we added a section about the machine learning (line 185-191). We explain how the algorithm classifies pixel/voxel using the random forest classifier. |
| 4. | Line 156, how did the authors calculate the error bound? | The error bound is one of the mandatory settings for the solver. Since every calculated permeability is a result of an iterative differential equation process, the "true" value for permeability can only be approximated. A low error bound value of |

| | | 0.05 often requires a simulation time of days to reach the specified stopping criterion. We added a description in the manuscript from line 164-167. |
|---|---|---|
| 5. | Line 258, modify the absolute permeability format. | We changed the format of the permeability to mD. |
| 6. | The unit of the permeability should be consistent for the whole paper | We made sure that the permeability format is consistent in the manuscript. |
| 7. | Additional question: Is this method applicable for shale? | Thank you for asking. Since the structure of shales is very different compared to sandstones, the method might be applicable depending on the pore throat characteristics. In shales, the dominating pore structure can be either fractured or porous while organic matter may play a role too (Tiwari et al. 2013, Grathoff et al. 2016). That is why an investigation of the method using mudrocks might be the most natural next step to climb. In the porous case, the method might be applicable when the pores are large enough while in the other cases the modelled clay mineral content might have no or only a minor effect on the permeability calculations. Further studies are necessary to gain knowledge about the effect of clay modelling on permeability in shales when our method should be applied. Of course, this is a very promising topic, and the method should be benchmarked on these structures as well. We added two sentences about the possible applicability in other types of rocks from line 351-354. |

| | Further changes in the manuscript |
|---|---|
| 1. | We changed the sentence about Darcy´s law in line 150 since the meaning was misleading. |
| 2. | We added a sentence about the validity of the flow field variances in line 318-319. |
| 3. | We added new references which emphasize important aspects of the manuscript. |

[revised manuscript text omitted]

---

## Author Response (AR2)

| | Saeid Sadeghnejad: | Authors' reply: |
|---|---|---|
| 1. | The paper's Appendix comparing micro-CT, BSE, and EDX should be merged into the main text. Fig A provides a clear insight into the "problem definition" of the paper. The readers should wait until reading this Appendix and see this figure to understand why Illites are not visible in micro-CT images. | As suggested by Mr. Sadeghnejad, we inserted the images into the manuscript text section (Figure 2,5). This improves the comprehensibility of the manuscript, clarifies the motivation, and illustrates why we had to model illites into µXCT structures to obtain the true rock permeability. |
| 2. | Authors considered a 2-voxel pore layer next to solid surfaces as pore locations for illites in the primary simulation model. The reason for selecting this value was not discussed in the paper. Also, the methodology used in Eq. 7 requires more detailed explanations. | We updated the description of the decision why we chose a 2-voxel layer of illites and included it into the text. We updated the description of the method in line 213-216. Furthermore, we updated the text from line 221-224. We changed fig. 6 and the figure description to improve the comprehensibility of the method. |
| 3. | However, the discussion of the same effect on the porosity of the system is missing. | With a focus on the permeability impact of clay minerals, this work shows that only a minor alteration of the pore throat morphology can be accompanied by a strong reduction of permeability in tight reservoir rocks. Assuming porous voxels, the porosity does not change significantly by applying the methodology. It is our intention to emphasize that the well-known relationship between porosity and permeability changes with a disconnection of the pore throats (e.g. Leu et al., 2014). The impact on permeability of this effect increases with decreasing permeability of the considered rock. |
| | Anonymous referee #1: | |
| 1. | Line 40-41, in the Introduction section, the following references (Liuand Mostaghimi 2018; Liu et al. 2020) can't be ignored to discuss the recent advances in the pore-scale studies below µXCT resolution. Liu, M., & Mostaghimi, P. (2018). Reactive transport modelling in dual porosity media. Chemical Engineering Science,190, 436-442. Liu, M., Starchenko, V., Anovitz, L. M., & Stack, A. G. (2020). Grain detachment and transport clogging during mineral dissolution in carbonate rocks with | We appreciate your comment and added the suggested references to line 40-42. |

| | | |
|---|---|---|
| | C1SED Interactive comment Printer-friendly version Discussion paper permeable grain boundaries. Geochimica et Cosmochimica Acta, 280, 202-220. | |
| 2. | Line 87, "The term....", this sentence seems have different font style with the context. Please check it through the paper. | We checked for different font styles and corrected it throughout the manuscript. |
| 3. | Line 89, reference is needed for the definition of "nanoporous". | We added a description for the term "nanoporous" from line 88-90 (e.g. Tinet et al. 2020). |
| 4. | Line 208, can the authors provide the values of the parameters used in Equation (1)? | Since line 208 does not contain information about the equation, we think you might have meant line 108. We added a section about the physical parameters which were used for the gas permeameter test (line 244-254). These include values of the applied differential pressure and size of the samples. Furthermore, we added an example for the applied pressures for sample F8 (Table 1). |
| 5. | Scale bar is missing for Figure 2. | In Figure 3 (former Figure 2), there is a scale bar in the bottom left corner of the figure. |
| 6. | Line 175, I think there should be more introduction about the machine learning method used in Figure 3. | *Ilastik* uses a Random Forest algorithm to provide different trees for the classification of the voxels/pixels. This is used to achieve a segmentation based on not only greyscale values, but shape as well. We added a section about the machine learning method from line 185-191. |
| | Anonymous referee #2: | |
| 1. | Line 104, list the brand, type and accuracy of the differential pressure transducer. | We listed the brand, type and accuracy of the equipment we have used from line 108-111. |
| 2. | Line 111, MIP should be replaced by MICP. | We replaced the term "MIP" by "MICP" throughout the text. |
| 3. | Line 128, add more details of the machine learning segmentation used in this study | Following the referees' suggestion, we added a section about the machine learning (line 185-191). We explain how the algorithm classifies pixel/voxel using the random forest classifier. |
| 4. | Line 156, how did the authors calculate the error bound? | The error bound is one of the mandatory settings for the solver. Since every calculated permeability is a result of an iterative differential equation process, the "true" value for permeability can only be approximated. A low error bound value of |

| | | |
|---|---|---|
| | | 0.05 often requires a simulation time of days to reach the specified stopping criterion. We added a description in the manuscript from line 164-167. |
| 5. | Line 258, modify the absolute permeability format. | We changed the format of the permeability to mD. |
| 6. | The unit of the permeability should be consistent for the whole paper | We made sure that the permeability format is consistent in the manuscript. |
| 7. | Additional question: Is this method applicable for shale? | Thank you for asking. Since the structure of shales is very different compared to sandstones, the method might be applicable depending on the pore throat characteristics. In shales, the dominating pore structure can be either fractured or porous while organic matter may play a role too (Tiwari et al. 2013, Grathoff et al. 2016). That is why an investigation of the method using mudrocks might be the most natural next step to climb. In the porous case, the method might be applicable when the pores are large enough while in the other cases the modelled clay mineral content might have no or only a minor effect on the permeability calculations. Further studies are necessary to gain knowledge about the effect of clay modelling on permeability in shales when our method should be applied. Of course, this is a very promising topic, and the method should be benchmarked on these structures as well. We added two sentences about the possible applicability in other types of rocks from line 351-354. |

| | Further changes in the manuscript |
|---|---|
| 1. | We changed the sentence about Darcy´s law in line 150 since the meaning was misleading. |
| 2. | We added a sentence about the validity of the flow field variances in line 318-319. |
| 3. | We added new references which emphasize important aspects of the manuscript. |

**Authors' reply to the editor:**

The authors want to thank the editor for the effort he spent evaluating our manuscript. We addressed the comments and statements of the editor in our manuscript (highlighted in green).

Dear Authors,

I have now had time to evaluate the reviews and read your manuscript in detail, apologies for the slight delay.

This is a very interesting, well-written and well-illustrated paper, and I look forward to seeing it published in SE. Before I will recommend it for publication and in order to ensure the reproducibility of your work, I would ask you to provide the numerical details of your segmentation choices - that is, for the uCT data, the threshold values and the bit depth of your data as well as details on the reconstruction algorithm you used. Given that the ML segmentation hinges on the training of the classifier, I would ask to make the classifier file you used for segmentation available as supplementary material. In order to evaluate the influence of phase contrast on your synchrotron data, it would be important that you provided the sample-detector distance.

Please also have a look at the attached, annotated pdf.

With kind regards,

Florian Fusseis

Non-public comments to the Author:

W/ regards to the references that referee #1 requests, please only include these if they are essential.

Authors' reply:

| | Changes in the manuscript |
|---|---|
| 1. | As suggested by the editor, we corrected line 32, 40, 54, 70-71, 81-82, 219, 220, 288, 322. |
| 2. | We added a section about the synchrotron-based μXCT scanning and provided additional information about thresholding values, sample-detector distance and the datatype of the images (127-139). We used one machine learning method for the qualitative comparison with conventional thresholding. We clarified this in the manuscript. |
| 3. | We provided the .ilp file which includes all information about the classifier. This file may be opened with a .h5 compatible program (MATLAB). To gain access with MATLAB, the command would be: hinfo = hdf5info('filename') |

| | An exemplary .tif file of a slice of the illite meshwork which can be opened along with the .ilp file was attached as well. The slice illustrated in Fig. 4 was used for this purpose (Step C to D). |
|---|---|
| 4. | We included the references provided by referee #1 since they helped to clarify our statements about combining FIB-SEM with μXCT. |

[revised manuscript text omitted]